# Effects of Magnesium on Transcriptome and Physicochemical Index of Tea Leaves

**DOI:** 10.3390/plants12091810

**Published:** 2023-04-28

**Authors:** Ying Zhang, Qi Zhang, Yuhua Wang, Shaoxiong Lin, Meihui Chen, Pengyuan Cheng, Yuchao Wang, Mengru Du, Xiaoli Jia, Haibin Wang, Jianghua Ye

**Affiliations:** 1College of Tea and Food, Wuyi University, Wuyishan 354300, China; 2College of Horticulture, Fujian Agriculture and Forestry University, Fuzhou 350002, China; 3College of Life Science, Fujian Agriculture and Forestry University, Fuzhou 350002, China; 4College of Life Science, Longyan University, Longyan 364012, China

**Keywords:** tea tree, magnesium, transcriptomics, photosynthesis, quality

## Abstract

Magnesium (Mg) is one of the essential elements for the growth of tea tree and is extremely important for its development. In this study, we investigated the effect of Mg on the transcriptome and physicochemical indexes of tea leaves, and the results showed that Mg could significantly affect the gene expression of tea leaves. The results of Orthogonal Partial Least-Squares Discriminant Analysis (OPLS-DA) model analysis showed that a total of 300 key genes (Variable Importance for the Projection, VIP > 1) were screened under different concentrations of Mg treatment, among which 140 genes were up-regulated and 160 genes were down-regulated. The bubble map was used to screen the characteristic genes from the above key genes, and a total of 121 representative characteristic genes were obtained, mainly involving 9 metabolic pathways. Among them, gene expression of three metabolic pathways, including porphyrin metabolism, alpha-linolenic acid metabolism and photosynthesis, showed an increasing trend with the increase of Mg concentration, while gene expression of four metabolic pathways, including biosynthesis of secondary metabolites, anthocyanin biosynthesis, ABC transporters, pentose and glucuronate interconversions, showed a decreasing trend. The results of physiological index analysis showed that with the increase of Mg concentration, the photosynthetic physiological index, theanine and soluble sugar content of tea leaves showed an increasing trend, while the content of tea polyphenol, flavone and caffeine showed a decreasing trend. The results of TOPSIS analysis showed that the physiological indexes of tea trees most affected by Mg were chlorophyll, tea polyphenols and flavonoids, while the metabolic pathways most affected by Mg on gene expression were the metabolic pathways and biosynthesis of secondary metabolites. It can be seen that the effects of Mg on tea tree were mainly related to photosynthesis and synthesis of secondary metabolites, and Mg was beneficial for improving the photosynthetic capacity of tea tree, enhancing the accumulation of primary metabolites, and thus increasing tea yield. However, Mg was not conducive to the synthesis of secondary metabolites of tea tree and the accumulation of main quality indexes of tea leaves.

## 1. Introduction

Magnesium (Mg), one of the 16 elements required for plant growth, is one of the most abundant macronutrients in plants and one of the most abundant cations in living cells [1]. In plants, Mg is present as ions or organic compounds and acts as a cofactor for a range of enzymes involved in various physicochemical processes, such as photosynthesis, respiratory metabolism, and nucleic acid metabolism [2]. For example, Mg can affect plant growth by activating DNA and RNA synthesis processes [3]. Mg is one of the synthetic components of chlorophyll synthesis, and Mg deficiency reduced the ability of chlorophyll synthesis and photosynthesis [4]. Mg activated various phosphate mutases and phosphokinases during plant respiration, which in turn affected the plant respiratory metabolic capacity [5]. Secondly, Mg can effectively improve the absorption of macronutrients by plants, which in turn promotes the absorption and utilization of nutrients by plants [6]. It can be seen that Mg is extremely important for plant growth, and that Mg deficiency can lead to lower productivity and yield, and effective regulation of Mg supply in the process of plant growth is of great significance for promoting plant growth.

The tea industry is an important agricultural industry in China and has made an important contribution to the development of Chinese agriculture. In 2022, the area of tea plantations in China reached 3.2 million hectares, and the total production of dry raw tea exceeded 3.2 million tons, with a total output value of more than RMB 300 billion. Tea trees are mainly harvested from young buds and leaves, and a large amount of fertilizer is needed to improve the yield of tea tree [7,8]. Therefore, numerous scholars have carried out many studies on the effects of fertilizers on tea tree growth, tea yield and quality, but have mainly focused on the macroelement elements required by tea trees, such as nitrogen, phosphorus and potassium fertilizers [9,10,11,12]. The requirement for Mg during plant growth is not as high as that for N, P and K, but Mg is extremely important for plant growth [13]. It has been reported that the appropriate use of magnesium fertilizer can promote the accumulation of nutrients in plant roots, promote plant root growth, and improve root transport capacity, thereby increasing plant biomass [14,15]. Second, Esteves et al. [16] found that Mg had a greater effect on soil and nutrient concentration than nitrogen fertilizer or calcium fertilizer. For tea trees, the use of Mg fertilizer could improve photosynthetic capacity, promote tea tree growth and increase tea yield [17]. The Mg content in tea leaves changes significantly under the interference of external factors. For example, Mg content in tea leaves increased significantly after pruning, and the yield of tea trees in the second year after pruning was higher than that of unpruned tea trees [18]. It can therefore be seen that Mg is extremely important in the growth of tea trees. However, there are few reports on how Mg affects tea tree growth, particularly in relation to how Mg regulates gene expression in tea trees, which in turn affects the metabolic capacity of tea trees and regulates tea tree growth. To further reveal the molecular and physiological mechanism of Mg on the growth of tea tree is of great significance for the rational use of Mg fertilizer in tea tree cultivation, so as to regulate the yield and quality of tea trees.

Accordingly, in this study, tea trees were planted using hydroponics, Mg concentration in the culture medium was adjusted for treatment and changes in the photosynthetic and physiological indexes of tea leaves were measured after treatment with different Mg concentrations. At the same time, tea leaves treated with different Mg concentrations were collected to determine the quality index content and transcriptome gene expression of tea leaves, which in turn analyzed the physiological and molecular mechanisms of tea trees in response to Mg regulation, so as to provide some guidance for the rational use of Mg fertilizer in tea plantations.

## 2. Results and Discussion

### 2.1. Transcriptomic Analysis of Tea Leaves under Mg Regulation

A total of 455,039,944 Mbp raw reads were obtained from 9 libraries after Illumina sequencing. After filtering, 446,629,368 Mbp clean reads were identified, and the percentage of clean reads relative to raw reads in each library was above 99.97% (Appendix A), indicating that the results could comprehensively and truly reflect the tea leaf transcriptome. Comparison efficiency analysis showed that the proportion of clean reads successfully matched with reference genomes was higher than 80% (Appendix A). It can be seen that the assembly of the reference genome was relatively complete, the species measured were consistent with the reference genome, there was no contamination in related experiments and the results could be used for further analysis.

Further analysis of the number of genes detected in tea leaves under Mg regulation showed (Figure 1A) that the total number of genes in tea leaves differed significantly after treatment with different Mg concentrations (*p* < 0.001). The results of the principal component analysis (Figure 1B) showed that two principal components could distinguish three samples, with the contribution rate of 53.8% for principal component 1 and 30.4% for principal component 2, and the total contribution rate of 84.2%. The results of gene expression analysis in tea leaves showed (Figure 1C) that there were significant differences (*p* < 0.001) in gene expression levels of tea leaves treated after different Mg concentrations. The results of the principal component analysis showed (Figure 1D) that three samples could be distinguished by two principal components, and the contribution rates of principal component 1 and principal component 2 were 55.1% and 32.0%, respectively, while the total contribution rate was 87.1%. It can be seen that the gene expression of tea leaves changed significantly under Mg regulation.

On this basis, this study further analyzed the effects of different Mg concentrations on the gene expression of tea leaves, and the results showed that compared with the control (M1), 2634 genes were up-regulated and 1530 genes were down-regulated after 0.4 mmol/L Mg treatment (M2) (Figure 1E); compared with M2, 1578 genes were up-regulated and 2595 genes were down-regulated after 0.8 mmol/L Mg treatment (M3) (Figure 1F). Secondly, with the increase of Mg concentration (M1~M3), a total of 520 genes showed an increasing trend in expression level and 469 genes showed a decreasing trend in expression level. It can be seen that the expression of different genes in tea leaves changed significantly after treatment with different Mg concentrations, and Mg regulation could significantly affect gene expression in tea leaves.

### 2.2. Screening of Key Genes in Tea Leaves under Magnesium Regulation

OPLS-DA can be used to model the correlation between gene expression and samples, and to screen for key genes that characterize sample variability by variable importance projection value (VIP value) [19]. Meanwhile, to test the reliability of the OPLS-DA model, a permutation test is usually used to validate the model and thus evaluate the accuracy of the model [20]. Based on the above analysis, this study found that a total of 989 genes had significant changes in expression levels after the treatment with different Mg concentrations, of which 520 genes showed an upward trend in expression levels and 469 genes showed a downward trend with the increase of Mg concentration (M1~M3). In order to screen and obtain the key genes that changed after the treatment with different Mg concentrations, the OPLS-DA model was used for analysis in this study, and the results showed (Figure 2A,D) that the R^2^Y fit value was 1 (*p* < 0.005) and the predictive Q^2^ value was 0.964 (*p* < 0.005) for the OPLS-DA model of the control (M1) and 0.4 mmol/L Mg-treated samples (M2). The R^2^Y value of the OPLS-DA model for M2 and 0.8 mmol/L Mg-treated (M3) samples was 1 (*p* < 0.005), and the predictive Q^2^ value was 0.981 (*p* < 0.005). It can be seen that the R^2^Y and Q^2^ values of both models reached significant levels, and the models had a good degree of fit and high reliability, which can be used for further analysis.

The results of the OPLS-DA scoring chart showed (Figure 2B,E) that the OPLS-DA model could effectively distinguish between M1, M2 and M3 samples. There were significant differences in gene expression levels between M1, M2 and M3. S-plot analysis showed (Figure 2C,F) that 530 key genes (VIP > 1) distinguished between M1 and M2, and 591 key genes (VIP >1) distinguished between M2 and M3. Further analysis revealed that there were 300 common key genes (VIP > 1) that differentiated M1, M2 and M3; 140 key genes showed an increasing trend in expression level with the increase of Mg concentration (M1~M3), while 160 key genes showed a decreasing trend in expression level (Figure 3).

### 2.3. Screening of Characteristic Genes from Key Genes in Tea Leaves under Magnesium Regulation

On the basis of the above analysis, this study further analyzed the expression levels of 300 key genes distinguishing samples of different Mg concentrations. The bubble characteristic map analysis showed (Figure 4A,B) that the expression levels of 121 characteristic genes accounted for more than 90% of the expression levels of 300 key genes. Among them, 51 characteristic genes showed an increasing trend with the increase of Mg concentration (Figure 4C), while 70 showed a decreasing trend (Figure 4D). GO enrichment of characteristic genes showed (Figure 5A) that 121 characteristic genes were primarily involved in 27 biological functions or pathways (*p* < 0.05). The enrichment results of the KEEG pathway of characteristic genes showed (Figure 5B) that the 121 characteristic genes were primarily involved in 42 metabolic pathways, of which 9 metabolic pathways were enriched to a significant level (*p* < 0.05), such as secondary metabolite biosynthesis, porphyrin metabolism, anthocyanin biosynthesis, metabolic pathways, carotenoid biosynthesis, ABC transporters, pentose and glucuronate interconversions, alpha-linolenic acid metabolism and photosynthesis.

Further analysis of gene expression levels in nine metabolic pathways showed (Figure 5C) that, with the increase of Mg concentration, the gene expression of porphyrin metabolism, alpha-linolenic acid metabolism and photosynthesis pathways showed an upward trend, and the gene expression in secondary metabolite biosynthesis, anthocyanin biosynthesis, ABC transporters, pentose and glucuronate interconversion pathways showed a downward trend, while the gene expression of metabolic pathways and carotenoid biosynthesis pathways were first decreased and then increased. Porphyrin metabolism was closely related to chlorophyll synthesis in plants, and the enhancement of porphyrin metabolism was beneficial to promote chlorophyll synthesis in plants, thus improving plant photosynthetic capacity [21,22,23]. Linolenic acid, the main product of alpha-linolenic acid metabolism, is one of the major fatty acids contained in plant membrane lipids. For example, the main fatty acids contained in the membrane lipids of plant photosynthetic membrane (Thylakoid membrane) are linolenic acid, and the increase of its content is beneficial to improving plant photosynthesis [24,25]. At the same time, the decomposition of linolenic acid is also beneficial to increasing the content of secondary metabolites of plants, especially jasmonic acid [26,27]. In addition, jasmonic acid was shown to be beneficial to inducing and enhancing the phenylpropane metabolic pathway and flavonoid metabolic pathway in plants, thereby increasing the content of secondary metabolites [28]. Anthocyanin, which is the product of the anthocyanin biosynthesis pathway, is mainly derived from the phenylpropane metabolism and flavonoid metabolism pathway, and is a secondary metabolite of plants [29]. Secondly, it has been reported that pentose and glucuronate interconversions were enhanced when plants were subjected to environmental stress, thus improving carbohydrate and energy metabolism and promoting secondary metabolites synthesis [30]. In addition, environmental stress could improve the expression of ABC transporters and the ability to transport secondary metabolites across membranes [31]. In this study, we found that gene expression of porphyrin metabolism and alpha-linolenic acid metabolism pathways in tea leaves were enhanced with the increase of Mg concentration, as well as gene expression of the photosynthesis pathway. It can be seen that Mg regulation was beneficial for improving the photosynthetic capacity of tea trees. Secondly, this study found that the expression of biosynthesis genes of secondary metabolite pathways in tea leaves was down-regulated with the increase of Mg concentration, and the expression of the anthocyanin biosynthesis gene was also down-regulated. Meanwhile, gene expression of pentose and glucuronate interconversions and ABC transporters related to secondary metabolite synthesis and transport showed a decreasing trend. It can be seen that increasing Mg concentration was conducive to enhancing the photosynthetic capacity of the tea tree. However, it was not conducive to the synthesis of secondary metabolites, especially those from the phenyl propane and flavonoid metabolic pathways.

### 2.4. Analysis of Photosynthetic Physiological Indexes and Quality Indexes of Tea Leaves

The aforementioned transcriptome analysis of tea leaves revealed that Mg regulation could improve the expression of photosynthesis-related genes and decrease the expression of the secondary metabolic pathway genes in tea trees, which in turn could improve the photosynthesis capacity of tea trees and decrease the synthesis capacity of secondary metabolites in tea trees. Accordingly, this study further analyzed the photosynthetic physiological indexes of tea trees under Mg regulation, and the results showed (Figure 6A) that with the increase of Mg concentration, the photosynthetic physiological indexes (Y(II), Fv/Fm, Fm, F0 and leaves) of tea leaves showed an increasing trend. The results of the quality index analysis showed (Figure 6B) that the content of theanine and soluble sugar in tea leaves showed an increasing trend with the increase of Mg concentration, while the content of tea polyphenol, flavone and caffeine showed a decreasing trend. Theanine and soluble sugar are the main metabolites of the tea tree, and enhanced photosynthetic capacity is conducive to the increase of theanine and soluble sugar content [32]. Zhang et al. [33] also found that magnesium regulation could increase theanine content in tea leaves. Tea polyphenol, flavone and caffeine are secondary metabolites of the tea tree, and their synthesis mainly occurs in the phenylpropane metabolic pathway, flavonoid metabolic pathway and alkaloid metabolic pathway [34,35]. It can be seen that increasing the supply of Mg was conducive to improving the photosynthetic capacity and accumulation of the primary metabolites of tea trees, but not conducive to the synthesis and accumulation of secondary metabolites. The results also validated previous findings in tea leaf transcriptomics.

### 2.5. Analysis of Interaction between Physiological Indexes and Gene Expression of Different Metabolic Pathways in Tea Leaves

The results of redundancy analysis (RDA) physiological indexes and different metabolic pathways of tea leaves showed (Figure 7A) that photosynthetic physiological indexes of tea leaves were closely related to soluble sugar and theanine content of tea leaves. Flavonoid, tea polyphenol and alkaloid contents of tea leaves were closely related to biosynthesis of secondary metabolites, ABC transporters, anthocyanin biosynthesis, pentose and glucuronate interconversions and the carotenoid biosynthesis pathway in tea leaves. The results of the correlation interaction network analysis showed that (Figure 7B,C) there were significant positive correlations between Mg concentration, photosynthetic physiological indexes, soluble sugar and theanine content and gene expression of porphyrin metabolism, alpha-linolenic acid metabolism and the photosynthesis pathway; Mg concentration was negatively correlated with flavonoids, tea polyphenols, alkaloids and gene expression of secondary metabolites, ABC transporters and the anthocyanin biosynthesis pathway. The TOPSIS method was used to further analyze the weight of the effects of Mg regulation on physiological indexes and metabolic pathway gene expression. Weight analysis of photosynthetic physiological indexes showed (Figure 7D) that Mg regulation had the greatest effect on chlorophyll content. The weight analysis results of the quality indexes showed (Figure 7E) that Mg regulation had the greatest influence on the content of tea polyphenols, followed by flavonoids. Weight analysis of metabolic pathway gene expression showed (Figure 7F) that Mg regulation had the greatest influence on the metabolic pathways and biosynthesis of secondary metabolic pathway gene expression, followed by photosynthesis. It can be seen that Mg regulation mainly affected the photosynthetic capacity and secondary metabolic capacity of tea trees, and the increase of Mg concentration was conducive to improving the photosynthetic capacity of tea trees, but not conducive to the synthesis of secondary metabolites of tea trees.

## 3. Materials and Methods

### 3.1. Field Experiment and Sample Collection

In this study, we used hydroponics to grow tea seedlings and analyzed the effect of different Mg concentrations on the transcriptome and physiological and biochemical indexes of tea leaves. The selected tea tree species was Wuyi Rougui (*Camellia sinensis*). The specific experimental method was to select one-year-old tea seedlings (35 cm in height and 0.3 cm in diameter) of uniform growth, wash them with water to remove root soil and pre-culture them in nutrient solution with pH 4.5 for 45 days to recover and grow normally. The formulation of tea tree hydroponic nutrient solution was configured according to the method of Sun et al. [36] and the nutrient solution mainly contained 125 μmol/L KNO_3_, 187.5 μmol/L (NH_4_)_2_SO_4_, 100 μmol/L KH_2_PO_4_, 25 μmol/L K_2_SO_4_. 100 μmol/L CaCl_2_, 100 μmol/LMgSO4, 16 μmol/L FeSO_4_ and 200 μmol/L AlSO_4_. Then, the tea seedlings were taken out, and the roots were rinsed with deionized water 3 times, and then the tea seedlings were transplanted into a nutrient solution containing different Mg concentrations for cultivation, with 3 replicates per treatment. The nutrient solution with different Mg concentrations was configured according to the above method, while Mg concentrations in the nutrient solution were adjusted to 0 mmol/L (M1), 0.4 mmol/L (M2) and 0.8 mmol/L (M3), and concentrations of other ions were consistent with the above formulation. Tea seedlings were transplanted into nutrient solutions with different Mg concentrations for 21 days, and the same nutrient solution was changed every 7 days. Tea seedlings were cultured in a greenhouse with 24 h of continuous ventilation and 12 h of light (8:00~20:00) per day, with a light intensity of 1500 lux, a temperature of 25 °C during the light period and 20 °C for the rest and a humidity maintained at 75% ± 5%. After 21 days of treatment with different Mg concentrations, photosynthetic physiological indexes of tea leaves (penultimate leaf) were determined. Then, tea leaves (penultimate leaf) were collected and immediately frozen in liquid nitrogen, and subsequently used to determine tea quality indexes and tea leaf transcriptome.

### 3.2. Determination of Photosynthetic Physiological Indexes in Tea Leaves

The photosynthetic physiological indexes of tea leaves were mainly determined by chlorophyll content and fluorescence parameters. The chlorophyll content of tea leaves was determined with a chlorophyll analyzer (SPAD-502 PLUS, Konica Minolta, Osaka, Japan) with 5 replicates per treatment. The fluorescence parameters of tea leaves were determined using a PAM-2500 chlorophyll fluorescence analyzer (WALZ, Nuremberg, Bavaria, Germany) with 5 replicates per treatment, and the indexes were minimal fluorescence (F0), maximal fluorescence (Fm), maximal quantum yield of PSII (Fv/Fm) and actual photochemical efficiency of PSII in the light (Y(II)). In the process of determination, tea leaves were acclimated to the dark environment for 30 min using a measurement light of less than 0.05 μmol·m^−2^·s^−1^ and a saturated pulse light of 8000 μmol·m^−2^·s^−1^.

### 3.3. Determination of Quality Indexes of Tea Leaves

The collected tea leaves were deactivated biologically at 105 °C for 15 min, dried at 80 °C to a constant weight, ground and passed through a 60-mesh sieve for the determination of quality indexes of tea leaves. The quality indexes of tea leaves were mainly determined by tea polyphenols, theanine, caffeine, flavone and soluble sugar, with 3 replicates per treatment. Tea polyphenol content was determined via folinol colorimetry according to the National Standard of the People’s Republic of China (GB/T 8313-2018 for the determination of tea polyphenols and catechins in tea) [37]. Briefly, 1 g of tea sample was weighed, 5 mL methanol solution with 70% solubility was added, extracted in a water bath at 70 °C for 10 min and centrifuged; 1 mL of supernatant was taken; 5 mL of folinol reagent was added and reacted for 5 min; 4 mL of Na_2_CO_3_ (7.5%) was added, fixed and left for 60 min; absorbance was measured at 765 nm; and gallic acid was used as the standard curve to quantify tea polyphenol content. Theanine content in tea was determined by high-performance liquid chromatography according to the national standard of the People’s Republic of China (GB/T 23193-2017 Determination of theanine in tea by high-performance liquid chromatography) [38]. Briefly, 1 g of tea sample was weighed, 100 mL of boiling distilled water was added and extracted using a water bath at 100 °C for 30 min, filtered and brought to a constant volume. The extract was filtered through a 0.45 μm membrane and determined via high-performance liquid chromatography. Caffeine content was determined via ultraviolet spectrophotometry according to the national standard of the People’s Republic of China (GB/T 8312 2013 Tea: caffeine determination) [39]. Briefly, 3 g of tea sample was weighed, added to 450 mL of boiling distilled water, extracted in a water bath at 100 °C for 45 min, filtered and brought to a constant volume; 10 mL of filtrate was taken, 4 mL of hydrochloric acid (0.01 mol/L) was added and volume fixed at 100 mL, left to stand and filtered; the absorbance was measured at 274 nm and caffeine was used as the standard curve to quantify the caffeine content. The content of flavonoids and soluble sugars was determined via the aluminum trichloride colorimetric method and anthrone colorimetric method, respectively, using the methods of Wang et al. [40].

### 3.4. Transcriptome Analysis of Tea Leaves

#### 3.4.1. RNA Extraction and RNA Sequencing

Tea leaves collected after treatment with different concentrations of Mg were used for RNA extraction, with three replicates per sample. Tea leaf RNA was extracted using TRI Reagent (Molecular Research Center, Cincinnati, OH, USA). RNA degradation and contamination were monitored on 1% agarose gels. RNA purity was checked using the NanoPhotometer^®^ spectrophotometer (IMPLEN, Westlake Village, CA, USA). RNA concentration was measured using a Qubit^®^ RNA Assay Kit in Qubit^®^2.0 Flurometer (Life Technologies, Foster City, CA, USA). RNA integrity was assessed using the RNA Nano 6000 Assay Kit of the Bioanalyzer 2100 system (Agilent Technologies, CA, USA).

The specific methods of transcriptome sequencing library preparation were: a total amount of 1 μg RNA per sample was used as input material for the RNA sample preparations. Sequencing libraries were generated using NEBNext^®^UltraTM RNA Library Prep Kit for Illumina^®^ (NEB, Ipswich, MA, USA) following the manufacturer’s recommendations, and index codes were added to attribute sequences to each sample. Briefly, mRNA was purified from total RNA using poly-T oligo-attached magnetic beads. Fragmentation was carried out using divalent cations at elevated temperatures in a NEBNext First Strand Synthesis Reaction Buffer (5X). The first strand of cDNA was synthesized using a random hexamer primer and M-MuLV Reverse Transcriptase (RNase H-). Second-strand cDNA synthesis was then performed using DNA Polymerase I and RNase H. The remaining overhangs were converted into blunt ends through exonuclease/polymerase activities. After adenylation of 3′ ends of DNA fragments, NEBNext Adaptor with hairpin loop structure were ligated to prepare for hybridization. To select cDNA fragments preferentially 250 to 300 bp in length, library fragments were purified with AMPure XP (Beckman Coulter, Beverly, MA, USA). Then, 3 μL USER Enzyme (NEB, USA) was used with size-selected, adaptor-ligated cDNA at 37°C for 15 min followed by 5 min at 95 °C before PCR. Then, PCR was performed with Phusion High-Fidelity DNApolymerase, Universal PCR primers and Index (X) primers. Finally, PCR products were purified (AMPure XP) and library quality was assessed on Agilent Bioanalyzer 2100.

The clustering of the index-coded samples was performed on a cBot Cluster Generation System using TruSeq PE Cluster Kit v3-cBot-HS (Illumia, San Diego, CA, USA) according to the manufacturer’s instructions. After cluster generation, library preparations were sequenced on an Illumina platform and 150 bp paired-end reads were generated.

#### 3.4.2. Transcriptome Data Analysis

Data quality control was performed by using fastp v 0.19.3 to filter the original data, mainly by removing reads with adapters; paired reads were removed when the N content in any sequencing reads exceeded 10% of the base number of reads; for any sequencing reads, when the number of low-quality (Q ≤ 20) bases contained in the reads exceeded 50% of the base number of reads, those paired reads were detected [41]. All subsequent analyses were based on clean reads. The reference genome was GCF_004153795.1_AHAU_CSS_1_genomic.fna.gz, and was downloaded from https://ftp.ncbi.nlm.nih.gov/genomes/all/GCF/004/153/795/GCF_004153795.1_AHAU_CSS_1/ (accessed on 11 March 2023) and its annotation files, using HISAT v2.1.0 to construct the index and compare clean reads to the reference genome [41]. New transcript prediction was performed using StringTie v1.3.4d [42]. StringTie applied a network streaming algorithm and optional de novo to splice transcripts, which allowed for more complete and accurate splicing of transcripts and faster splicing compared to Cufflinks and other software. FeatureCounts v1.6.2/StringTie v1.3.4d was used to calculate gene alignment and FPKM [43]. FPKM is the most commonly used method for estimating gene expression levels. DESeq2 v1.22.1/edgeR v3.24.3 was used to analyze the differential expression between the two groups, and the *p* value was corrected using the Benjamini–Hochberg method [44]. The corrected *p* value and |log_2_foldchange| were used as the threshold for significant difference expression. Enrichment analysis of differential gene expression was based on the hypergeometric test [45]. For KEGG, the hypergeometric distribution test was performed using pathway unit; for GO, the GO term was used.

### 3.5. Statistical Analysis

Excel 2017 software was used to calculate the mean value and variance of the data; Rstudio 3.3 software was used to make cloud and rain maps, principal component maps, volcanic map, OPLS-DA simulation, bubble map, redundancy analysis and TOPSIS analysis [46]. Heml 1.0 software was used to make heat maps. Cytoscape_v3.9.1 software was used to make the correlation analysis chart.

## 4. Conclusions

In this study, we analyzed the effects of Mg on the transcriptome and physicochemical indexes of tea leaves, and the results showed (Figure 8) that Mg could significantly affect the gene expression and physiological characteristics of tea leaves, which were mainly manifested in two aspects. On the one hand, with the increase of Mg concentration, the expression of photosynthesis, porphyrin metabolism and alpha-linolenic acid metabolism genes in tea leaves were up-regulated, and the content of soluble sugar and theanine in tea leaves was also improved, which in turn enhanced the primary metabolic capacity of tea leaves and was conducive to the improvement of tea yield. On the other hand, with the increase of Mg concentration, the expressions of biosynthesis of secondary metabolites and anthocyanin biosynthesis genes in tea leaves were down-regulated, and the secondary metabolic capacity of tea leaves was reduced, which in turn led to the decrease in the content of secondary metabolites, such as tea polyphenols, flavonoids and caffeine in tea leaves, which was not conducive to the improvement of tea quality. It can be seen that Mg was extremely important for the growth of the tea tree, and its effect on tea trees was mainly related to photosynthesis and synthesis of secondary metabolites. Mg regulation was conducive to improving the photosynthetic capacity of tea trees, enhancing the accumulation of primary metabolites and thus increasing tea yield, but was not conducive to the synthesis of secondary metabolites of tea trees and the accumulation of main quality indicators of tea leaves. Therefore, when using Mg fertilizer to regulate the growth of tea trees, its usage should be controlled to ensure the yield and quality of tea leaves.

## Figures and Tables

**Figure 1 plants-12-01810-f001:**
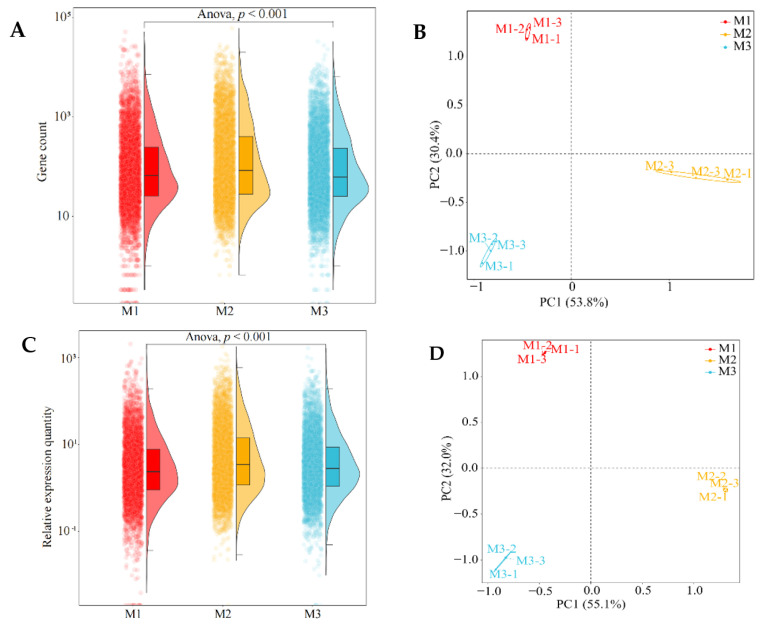
**Transcriptomic analysis of tea leaves under Mg regulation. Note: M1**: Control; **M2**: 0.4 mmol/L of Mg concentration; **M3**: 0.8 mmol/L of Mg concentration; (**A**) analysis of the total number of all genes detected in tea leaves treated with different Mg concentrations; (**B**) principal component analysis of gene quantity in tea leaves treated with different Mg concentrations; (**C**) analysis of the overall expression levels of all genes detected in tea leaves treated with different Mg concentrations; (**D**) principal component analysis of gene expression in tea leaves treated with different Mg concentrations; (**E**) volcanic map analysis of gene expression differences between M2 and M1; (**F**) volcanic map analysis of gene expression differences between M3 and M2.

**Figure 2 plants-12-01810-f002:**
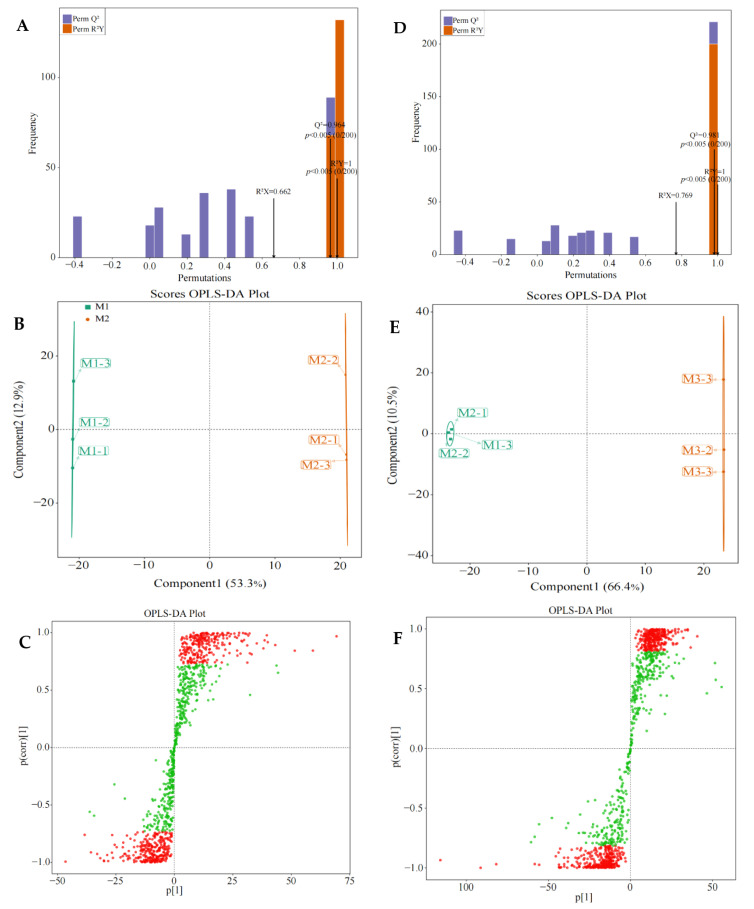
Screening of key genes in tea leaves treated with different Mg concentrations by OPLS-DA model. **Note: M1**: Control; **M2**: 0.4 mmol/L of Mg concentration; **M3**: 0.8 mmol/L of Mg concentration; (**A**) the fitting degree of M2 and M1 OPLS-DA model was tested to analyze the fitting degree and reliability of the model; (**B**) differentiation analysis of M2 and M1 using OPLS-DA model; (**C**) key differential genes of M2 and M1 screened using OPLS-DA model, and red dots represent the key differential genes; (**D**) the fitting degree and reliability of M3 and M2 OPLS-DA model; (**E**) differentiation analysis of M3 and M3 using OPLS-DA model; (**F**) the key differential genes of M3 and M2 screened using OPLS-DA model, and the red dots represent the key differential genes.

**Figure 3 plants-12-01810-f003:**
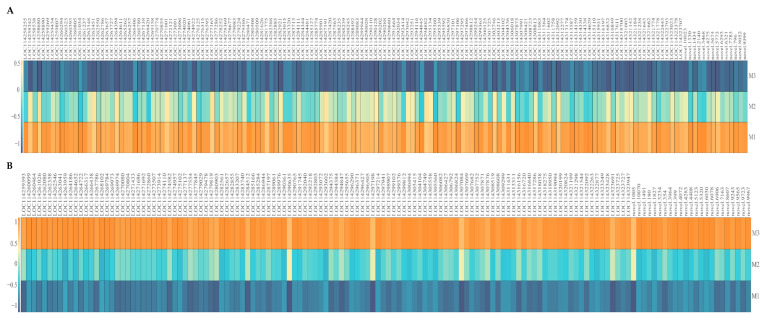
**Expression analysis of key genes screened by OPLS-DA model. Note: M1**: Control; **M2**: 0.4 mmol/L of Mg concentration; **M3**: 0.8 mmol/L of Mg concentration; (**A**) heat maps of key genes with up-regulated expression with the increase of Mg concentration; (**B**) heat maps of key genes with down-regulated expression with the increase of Mg concentration.

**Figure 4 plants-12-01810-f004:**
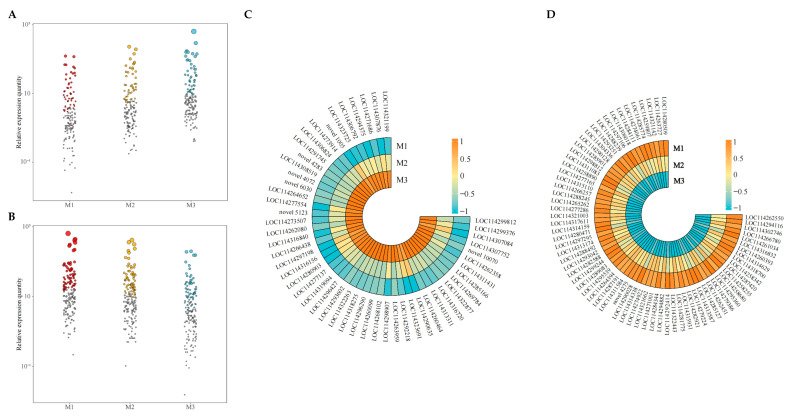
**Screening of characteristic genes with more than 90% key gene expression under magnesium regulation. Note: M1**: Control; **M2**: 0.4 mmol/L of Mg concentration; **M3**: 0.8 mmol/L of Mg concentration; (**A**) bubble characteristics of key genes with up-regulated expression after treatment with different Mg concentrations; (**B**) bubble characteristics of key genes with down-regulated expression after treatment with different Mg concentrations; (**C**) heat maps of characteristic genes with up-regulated expression with the increase of Mg concentrations; (**D**) heat maps of characteristic genes with down-regulated expression with the increase of Mg concentrations.

**Figure 5 plants-12-01810-f005:**
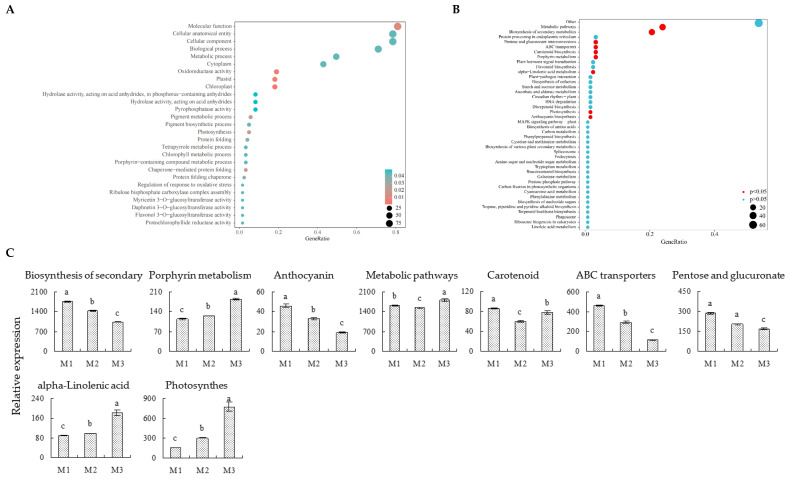
KEGG Pathway enrichment and gene expression analysis of characteristic genes. **Note: M1**: Control; **M2**: 0.4 mmol/L of Mg concentration; **M3**: 0.8 mmol/L of Mg concentration; (**A**) GO enrichment of characteristic genes; (**B**) KEGG pathway enrichment of characteristic genes; (**C**) total gene expression analysis of the significantly enriched KEGG pathway (error bars indicate standard error); Lowercase letters indicate significant differences at *p* < 0.05 levels.

**Figure 6 plants-12-01810-f006:**
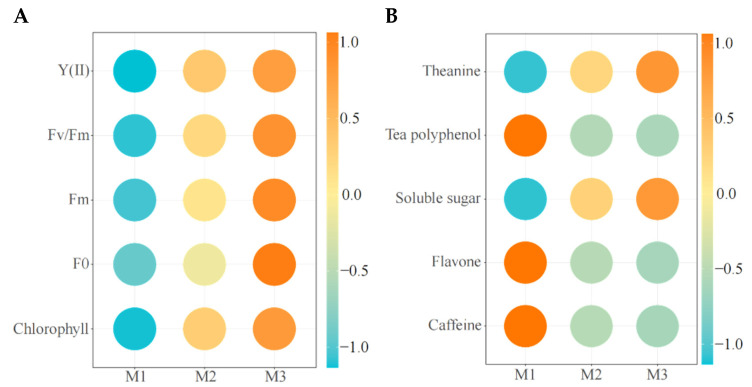
Physicochemical analysis of tea leaves under Mg regulation. **Note: M1**: Control; **M2**: 0.4 mmol/L of Mg concentration; **M3**: 0.8 mmol/L of Mg concentration; **Y(Ⅱ)**: actual photochemical efficiency of PSⅡ in the light; **Fv/Fm**: maximal quantum yield of PSⅡ; **Fm**: maximal fluorescence; **F0**: minimal fluorescence; (**A**) analysis of photosynthetic physiological indexes of tea leaves treated with different Mg concentrations; (**B**) analysis of quality indexes of tea leaves treated with different Mg concentrations.

**Figure 7 plants-12-01810-f007:**
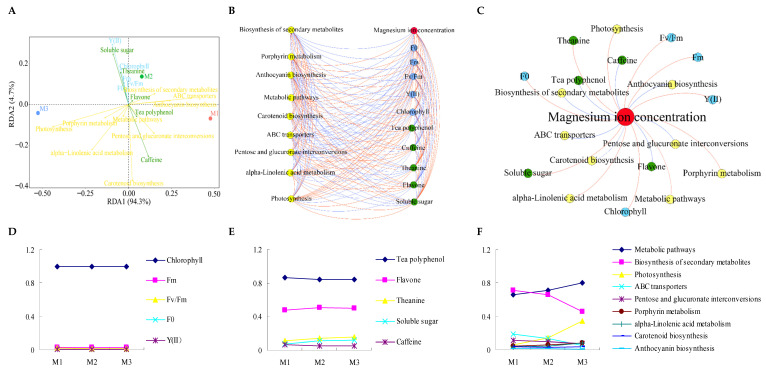
**Analysis of the relationship between physicochemical indexes and metabolic pathway gene expression intensity in tea leaves under Mg regulation and weight analysis of each index**. **Note: M1**: Control; **M2**: 0.4 mmol/L of Mg concentration is; **M3**: 0.8 mmol/L of Mg concentration; **Y(II)**: actual photochemical efficiency of PSII in the light; **Fv/Fm**: maximal quantum yield of PSII; **Fm**: maximal fluorescence; **F0**: minimal fluorescence; (**A**) redundancy analysis of physicochemical indexes and metabolic pathways in tea leaves; (**B**) interaction network analysis of physicochemical indexes and metabolic pathways in tea leaves, where the red line represents a positive correlation and the blue line is the negative correlation; (**C**) interaction network analysis between Mg ions and different indexes, where the red line represents a positive correlation and the blue line is the negative correlation; (**D**) weight analysis of photosynthetic physiological index; (**E**): weight analysis of quality indexes; (**F**): weight analysis of metabolic pathways.

**Figure 8 plants-12-01810-f008:**
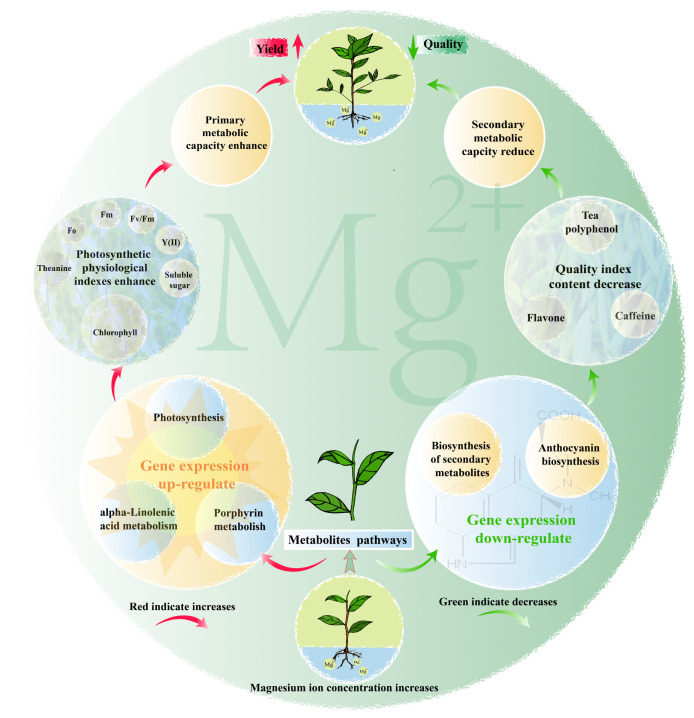
Mechanism diagram of the effect of magnesium regulation on tea tree growth.

## Data Availability

The data presented in this study are available as Appendix A. The original contributions presented in the study are publicly available. These data can be found here: NCBI, PRJNA944807.

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
