# Peer review of "Effects of Magnesium on Transcriptome and Physicochemical Index of Tea Leaves"

_plants, 2023, doi:10.3390/plants12091810_

Round 1

Reviewer 1 Report

The manuscript “Effects of magnesium on transcriptome and physiological and biochemical index of tea leaves” present the analysis of the transcriptome of Camellia sinensis variety Wuyi Rougui grown in hydroponics in response to different concentrations of Magnesium. The authors present some interesting results concerning the apparent antagonist effect of Mg on plant growth and secondary metabolism. However, the manuscript has several issues that need to be revised before being published, some of them are listed below:

Abstract

OPLS-DA and VIP are acronyms that should be explained.

Introduction

- The sentence: “For example, Mg could affect plant growth by activating DNA and RNA synthesis processes [3].” was not supported by the refereed paper.

– The text between lines 62-82 is repeated more than once and needs to be clarified.

Results

Line 99 – How could the authors speculate about the quality of the reference genome assembly?

Figure 1 – Charts 1A and 1C are incomprehensible and the captions do not clarify their meaning.

The authors should explain the advantages of the use of OPLS-DA as well as the reasons behind the use of PCA and OPLS-DA.

Figure 2 – The explanation of Figure 2 is mandatory that should include a brief explanation of the method of analysis.

Figure 3 – This figure does not fit on the A4 page.

Figure 4 – Caption is missed.

Figure 5C - Needs to be well explained, namely which genes contribute to the “relative expression quantity”.

The expression values obtained by NGS should be validated by RT-qPCR

Material and methods

The species name must be added.

The authors should verify if the tea quality indexes were analyzed after deep freezing.

Line 335 – Please rephrase the sentence “the collected tea leaves were killed” ?????

The methods used should be completely explained since the references are all in Chinese.

Line 408 – The Benjamini & Hockberg method needs a reference.

The software used in the statistical analysis needs a reference and/or the webs site.

Author Response

Reviewer 1

The manuscript “Effects of magnesium on transcriptome and physiological and biochemical index of tea leaves” present the analysis of the transcriptome of Camellia sinensis variety Wuyi Rougui grown in hydroponics in response to different concentrations of Magnesium. The authors present some interesting results concerning the apparent antagonist effect of Mg on plant growth and secondary metabolism. However, the manuscript has several issues that need to be revised before being published, some of them are listed below:

Abstract

OPLS-DA and VIP are acronyms that should be explained.

Answer: Thanks to the reviewer, the author has added.

Introduction

- The sentence: “For example, Mg could affect plant growth by activating DNA and RNA synthesis processes [3].” was not supported by the refereed paper.

Answer: Thanks to the reviewer. The author has revised it.

– The text between lines 62-82 is repeated more than once and needs to be clarified.

Answer: Thanks to the reviewer. The author has revised it.

Results

 Line 99 – How could the authors speculate about the quality of the reference genome assembly?

Answer: Thanks to the reviewer. Comparison efficiency refers to the percentage of Mapped Reads to Clean Reads, and is the most direct reflection of the utilization rate of transcriptome data. If the assembly of the reference genome is relatively perfect, and the species tested are consistent with the reference genome, and there is no contamination in related experiments, then the proportion of sequenced reads generated by the experiment that successfully match to the genome will be higher than 70% (Total Mapped).. In this study, the comparison efficiency reached more than 80%, indicating that the assembly quality of the reference genome was relatively perfect. This result is described in 2.1 of the results and discussion in the manusript.

The reference genome used in this study is GCF_004153795.1_AHAU_CSS_1_genomic.fna.gz.

Download address: https://ftp.ncbi.nlm.nih.gov/genomes/all/GCF/004/153/795/GCF_004153795.1_AHAU_CSS_1/.

Figure 1 – Charts 1A and 1C are incomprehensible and the captions do not clarify their meaning.

Answer: Thanks to the reviewer, the author has added.

The authors should explain the advantages of the use of OPLS-DA as well as the reasons behind the use of PCA and OPLS-DA.

Answer: Thanks to the reviewer. Principal component analysis (PCA) is a commonly used dimensionality reduction algorithm for unsupervised learning. OPLS-DA is a further supervised statistical method for discriminant analysis than PCA. The application and advantages of OPLS-DA have been highlighted in the results and discussion section of this paper.

Figure 2 – The explanation of Figure 2 is mandatory that should include a brief explanation of the method of analysis.

Answer: Thanks to the reviewer. The author has made some appropriate additions in the article.

Figure 3 – This figure does not fit on the A4 page.

Answer: Thanks to the reviewer. The author temporarily put this part in the paper, the author will follow the editor's requirements, if not appropriate, we can put it in the supplementary materials as an attachment.

Figure 4 – Caption is missed.

Answer: Thanks to the reviewer. The author checks and finds that there is a title in Figure 4, which is not shown, perhaps due to typography. The author has adjusted the format of the Figure.

Figure 5C - Needs to be well explained, namely which genes contribute to the “relative expression quantity”.

Answer: Thanks to the reviewer. The genes used for pathway enrichment in Figure 5 are all genes with significant differentially expressed genes after multi-level screening by volcano map, OPLS-DA, bubble characteristic map, etc. The up-regulated and down-regulated genes have been shown graphically in this paper. The authors did not list all the different genes because there are many of them.

The expression values obtained by NGS should be validated by RT-qPCR

 Answer: Thanks to the reviewer. In this study, the analysis was carried out from the overall level of the transcriptome. During the analysis process, the number of genes presenting differential expression was very large. Usually, the metabolic pathways with significant changes in gene expression could only be found through the enrichment of metabolic pathways and the overall analysis was carried out. Gene expression and RT-qPCR involving special pathways take a lot of time to complete, and we are also studying this aspect. Thank you very much for your advice.

Material and methods

The species name must be added.

Answer: Thanks to the reviewer, the author has added.

The authors should verify if the tea quality indexes were analyzed after deep freezing.

Answer: Thanks to the reviewer. Tea leaves were stored in liquid nitrogen immediately after sampling for subsequent index determination. The author describes in the Material Method.

Line 335 – Please rephrase the sentence “the collected tea leaves were killed” ?????

Answer: Thanks to the reviewer. The author has revised it.

The methods used should be completely explained since the references are all in Chinese.

Answer: Thanks to the reviewer, the author has added.

Line 408 – The Benjamini & Hockberg method needs a reference.

Answer: Thanks to the reviewer. The Benjamini-Hockberg method is the name of a method, also commonly called the Multiple comparison correction method. This method was proposed by Benjamini and Hochberg in 1995. In order to commemorate their contribution, scholars usually call it Benjamini-Hockberg method directly when using this method.

The software used in the statistical analysis needs a reference and/or the webs site.

Answer: Thanks to the reviewer. The software used for data analysis in this study, such as Excel 2017, Rstudio 3.3, Heml 1.0, and Cytoscape_v3.9.1, are all general professional analysis software that are available for free on the Web.

Reviewer 2 Report

.

First of all, I noticed that the authors stated "it has not been reported how Mg affects the growth of tea tree, especially how Mg regulates gene expression whin tea tree, which in turn affects the metabolic capacity of tea tree and regulates the growth of tea tree". However, Zhang et al. (2023) recently published a paper on the effect of Mg on the growth of tea plant (https://doi.org/10.1093/plphys/kiad143 ). Unfortunately, despite all the effort and hard work of the authors, I am not convinced about the novelty in the manuscript.

Moreover, the presented results are formulated very widely (such as "gene expressions of four metabolic pathways including biosynthesis of secondary metabolites, anthocyanin biosynthesis, ABC transporters, pentose and glucuronate interconversions, showed a decreasing trend"). 

The results, especially those of transcriptomic analyses, are described very descriptively without deeper biological interpretation.

Author Response

Reviewer 2

Comments and Suggestions for Authors

First of all, I noticed that the authors stated "it has not been reported how Mg affects the growth of tea tree, especially how Mg regulates gene expression whin tea tree, which in turn affects the metabolic capacity of tea tree and regulates the growth of tea tree". However, Zhang et al. (2023) recently published a paper on the effect of Mg on the growth of tea plant (https://doi.org/10.1093/plphys/kiad143 ). Unfortunately, despite all the effort and hard work of the authors, I am not convinced about the novelty in the manuscript.

Moreover, the presented results are formulated very widely (such as "gene expressions of four metabolic pathways including biosynthesis of secondary metabolites, anthocyanin biosynthesis, ABC transporters, pentose and glucuronate interconversions, showed a decreasing trend"). 

The results, especially those of transcriptomic analyses, are described very descriptively without deeper biological interpretation.

Answer: Thanks to the reviewer. The article of Zhang et al.(2023) was published online on March 4, 2023, and mainly discussed the effect of magnesium on nitrogen metabolism in tea trees, with the key focus on the relationship between magnesium and nitrogen and its effects on tea yield (mainly measuring biomass) and quality (mainly measuring theanine and free amino acids content). The study of Zhang et al.(2023) found that magnesium regulation was beneficial in increasing the biomass of tea trees and increasing the theanine and free amino acids content in tea leaves, and they believed that magnesium regulation could improve the quality of tea leaves.

       However, our study investigated the effect of magnesium regulation on tea tree growth and tea quality from the perspective of overall gene expression in the transcriptome, and found that magnesium regulation did promote the enhancement of photosynthetic capacity of tea tree, which indirectly supported the conclusion of Zhang et al.(2023) that magnesium regulation could increase tea tree biomass. Secondly, it was found in our study that magnesium regulation did increase theanine content in tea leaves, which supported the conclusion of Zhang et al.(2023). Therefore, the authors have added the article of Zhang et al.(2023) as a reference to the results and discussion. It is well known that the important indexes for evaluating the quality of tea are not only theanine and free amino acids content, but also tea polyphenols, caffeine, flavonoids content and so on. In our study, the metabolic pathways involved in key differentially expressed genes and quality indexes in tea leaves were determined under magnesium regulation, and it was found that magnesium regulation did increase the theanine content in tea leaves, but not conducive to the synthesis and accumulation of tea polyphenols, caffeine and flavonoids. The results of our study confirm the conclusion of the study of Zhang et al.(2023), and also contains important new discoveries that magnesium regulation was suitable for increasing tea yield, but there were great differences in tea quality indexes.

        Secondly, in our study, the significance of up-regulated or down-regulated gene expression of different metabolic pathways has been elaborated in the results and discussion, and the relationship between these pathways and tea tree growth and tea quality.

        In conclusion, our study is of great significance compared with previous studies, which not only validates the conclusions of previous studies, but also finds important results. Thanks again for the reviewer's suggestions.

Reviewer 3 Report

The first sentence in the Abstract should be correct, e,g.  Magnesium (Mg) is one of the essential elements for the growth of tea trees and is extremely important for its development.

The authors should explain why they used such concentrations of magnesium for fortifications.

Unfortunately, some figures were not available to read due to their horizontal location.

Author Response

Reviewer 3

Comments and Suggestions for Authors

The first sentence in the Abstract should be correct, e,g.  Magnesium (Mg) is one of the essential elements for the growth of tea trees and is extremely important for its development.

Answer: Thanks to the reviewer. The author has revised it.

The authors should explain why they used such concentrations of magnesium for fortifications.

 Answer: Thanks to the reviewer. I am very sorry that the author clearly wrote that the concentration of magnesium was mmoL/L in the material method. Due to the author's negligence, it was mistakenly written as moL/L in the follow-up. The author has carefully reviewed and revised all of them.

Unfortunately, some figures were not available to read due to their horizontal location.

Answer: Thanks to the reviewer. The author has adjusted the format of the figure.

Round 2

Reviewer 2 Report

I really appreciate the authors' effort to improve their work and take into account the suggestions. 

I have several comments to the presented manuscript:

Please, cite all the bioinformatic/statistical tools/algorithms (subsection 3.4.2, 3.5). Please, rephrase the transcriptomic analysis description using the same language style as in the other parts. (whole sentences, not just past participle). 

Fig. 5C Please, explain the error bars. Do the column represent the total transcript abundance of genes associated with the mentioned KEGG pathways? (If so, it should be briefly explained in the caption).

Fig. 8 Please, improve the figure resolution. It is difficult to read the small letters. The authors should also consider to simplify the scheme representation (one colour background instead of linear gradient). 

Author Response

I really appreciate the authors' effort to improve their work and take into account the suggestions.

I have several comments to the presented manuscript:

Please, cite all the bioinformatic/statistical tools/algorithms (subsection 3.4.2, 3.5).  Please, rephrase the transcriptomic analysis description using the same language style as in the other parts.  (whole sentences, not just past participle).

Answer: Thanks to the reviewer. The authors added references and revised the English description

Fig. 5C Please, explain the error bars. Do the column represent the total transcript abundance of genes associated with the mentioned KEGG pathways?  (If so, it should be briefly explained in the caption).

Answer: Thanks to the reviewer, the author has added.

Fig. 8 Please, improve the figure resolution.  It is difficult to read the small letters.  The authors should also consider to simplify the scheme representation (one colour background instead of linear gradient).

Answer: Thanks to the reviewer. The author replaced it with a clearer picture.